



# Laser pulse bidirectional reflectance from CALIPSO mission

Xiaomei Lu [1,2], Yongxiang Hu [2], Yuekui Yang [3], Mark Vaughan [2], Zhaoyan Liu [2], Sharon Rodier [1,2], William Hunt [1,2], Kathy Powell [2], Patricia Lucker [1,2], Charles Trepte [2]

[1] Science Systems and Applications, Inc., Hampton, VA, 23666, USA
5    [2] NASA Langley Research Center, Hampton, VA, 23681, USA
[3] NASA Goddard Space Flight Center, Greenbelt, MD, USA

*Correspondence to*: Xiaomei Lu (xiaomei.lu@nasa.gov); Yongxiang Hu (yongxiang.hu-1@nasa.gov)

**Abstract.** This paper presents an innovative retrieval method that translates the CALIOP land surface laser pulse returns into the surface bidirectional reflectance. To better analyze the surface returns, the CALIOP receiver impulse response and the 10    downlinked samples' distribution at 30 m resolution are discussed. The saturated laser pulse magnitudes from snow and ice surfaces are recovered based on surface tail information. The retrieved snow surface bidirectional reflectance is compared with reflectance from both CALIOP cloud cover regions and MODIS BRDF/Albedo model parameters. In addition to the surface bidirectional reflectance, the column top-of-atmosphere bidirectional reflectances are calculated from the CALIOP lidar background data and compared with the bidirectional reflectances derived from WFC radiance measurements. The 15    retrieved CALIOP surface bidirectional reflectance and column top-of-atmosphere bidirectional reflectance results provide unique information to complement existing MODIS standard data products and are expected to have valuable applications for modellers.

## 1 Introduction

The surface reflectivity over the snow and ice covered polar regions is a crucial factor in the Earth's radiation budget 20    (Wiscombe and Warren, 1980). Although passive sensors, such as the Moderate-resolution Imaging Spectroradiometer (MODIS) instruments, which are operating on both Terra and Aqua spacecraft (Savtchenko et al., 2003), are being used by scientists from a variety of disciplines, including oceanography (McClain, 2009; NASA Goddard Space Flight Center, 2014a; NASA Goddard Space Flight Center, 2014b), biology (NASA Goddard Space Flight Center, 2014c) and atmospheric science (Platnick et al. 2015), their reflectance observations are limited to daylight segments of each orbit, and have large 25    uncertainties due to the low solar angles in the polar regions (Behrenfeld et al., 2013; Lu et al., 2017). In this regard, active sensors can complement passive sensors and provide new measurement opportunities and insights (Yang et al., 2013).

The Cloud-Aerosol Lidar and Infrared Pathfinder Satellite Observations (CALIPSO) mission builds on the experience of the Lidar In-space Technology Experiment (LITE), which flew a three-wavelength lidar on the Space Shuttle in 1994 (Hunt et al., 2009; Winker et al., 1996). The CALIPSO instrument suite consists of the Cloud-Aerosol Lidar with Orthogonal 30    Polarization (CALIOP) a two-wavelength (1064 nm, 532 nm) polarization sensitive lidar (532 nm only), the Infrared





Imaging Radiometer (IIR) that has three channels in the thermal infrared, and the Wide Field Camera (WFC) with a single channel centered at 650 nm. Additional details can be found in the CALPSO algorithm theoretical basis documents (ATBDs), which are available online at (https://www-calipso.larc.nasa.gov/resources/project_documentation.php). Since routine science operations began on 13 June 2006, data have been collected almost continuously. The CALIOP lidar

provides vertical profiles of elastic backscattering from a near nadir-viewing angle during both day and night between 82ºN and 82ºS (Hunt et al., 2009; Winker et al., 2009). Though largely ignored prior to launch, the signals from CALIOP surface returns have since been mined for a wealth of unanticipated new discoveries (Behrenfeld et al., 2017; He et al., 2016; Lu et al., 2016; Venkata and Reagan, 2016). These valuable signals provide unique information to complement existing A-Train Terra and Aqua MODIS passive remote sensing data products, including nighttime measurements, measurements underneath

aerosols and non-opaque clouds, measurements in polar regions during all seasons and over sea-ice, and direct measurements of depolarization ratio at 532 nm.

In this study, we focus on the retrieval of surface laser pulse bidirectional reflectance using the CALIOP 532 nm polarization channels and provide an initial assessment of the algorithm's performance. To better analyze the surface returns, the CALIOP receiver impulse response and the downlinked attenuated backscatter signal at 30 m resolution are discussed in

detail. The paper is organized as follows. The CALIPSO, MODIS and Ice, Cloud and land Elevation Satellite (ICESat) data used in this study are briefly introduced in section 2. In section 3, the CALIOP surface bidirectional reflectance retrieval technique is described and its performance is evaluated relative to other satellite-derived data products. Section 4 presents the column bidirectional reflectance retrieval and highlights some initial results. Finally, the conclusions drawn from our analyses are given in section 5.

**2 Data products**

In this section, we describe the data products that are used in this paper. These include the CALIOP version 4 data, CALIPSO WFC data, MODIS bidirectional reflectance distribution function (BRDF) and albedo products, and ICESat Geoscience Laser Altimetry System (GLAS) surface elevation data.

**2.1 CALIPSO data**

The CALIOP version 4.1 (V4.1) level 1 (L1) data product is used here to derive the surface integrated attenuated backscatter from 30 meters above to 300 meters below the surface, which is a critically important parameter used in the surface bidirectional reflectance retrieval described in Section 3. Compared with version 3 (V3) data, the CALIOP V4.1 data product substantially improves both the 532 nm and 1064 nm calibration accuracies (Getzewich et al., 2018; Kar et al., 2017; Vaughan et al., 2018). Changes to the calibration techniques were firmly rooted by a thoroughly documented and peer-

reviewed approach (Getzewich et al., 2016; Getzewich et al., 2018; Kar et al., 2017;  Vaughan et al., 2018). The product is available beginning with data from 13 June, 2006. More information about the data products including data availability, user



documentation, quality statements, sample read software, and tools for working with the data, etc. can be found at the Atmospheric Science Data Center (ASDC) web site (https://eosweb.larc.nasa.gov/project/calipso/calipso_table).

To obtain the surface laser pulse reflectance under cloudless skies, the integrated attenuated backscatter (IAB) above the surface is used to filter out signals with any significant atmospheric attenuation. The typical clear atmosphere is considered

to be the case with IAB less than 0.0125 sr$^{-1}$ (Venkata and Reagan, 2016). The CALIOP level 2 (L2) data products (Vaughan et al., 2017) provide estimates of cloud/aerosol optical properties, such as layer optical depths, cloud/aerosol layer base and top altitudes, IAB above the Earth's surface, etc.

The WFC is a commercial-off-the-shelf instrument based on the Ball Aerospace & Technologies Corporation CT-633 star tracker design. The IFOV of each pixel in the camera is approximately 125 m by 125 m when projected onto the Earth's

surface from a nominal 705 km orbit. The image plane is oriented such that the active row of pixels is aligned in the cross-track direction providing a full swath FOV of approximately 61 km in the cross-track direction centered on the lidar boresight. The WFC is designed to have a large dynamic range that will allow it to observe bright clouds without saturation and still be able to detect small variations in surface albedo (Pitts et al., 2005). The bi-directional reflectance from CALPSO WFC radiance measurements is archived in WFC L1 data. The CALIOP L1, L2 and WFC L1 data products can be freely

downloaded         through         the         ASDC         at         NASA         Langley         Research         Center (https://eosweb.larc.nasa.gov/HORDERBIN/HTML_Start.cgi).

## 2.2 MODIS BRDF/Albedo Parameters

MODIS is an optical scanner that measures Earth radiance in 36 bands, ranging from 0.4 to 14 μm. The MODIS land surface reflectance is a seven-band product computed from the MODIS Level 1B land bands 1 (620-670 nm), 2 (841-876 nm), 3

(459-479 nm), 4 (545-565 nm), 5 (1230-1250 nm), 6 (1628-1652 nm) and 7 (2105-2155). The product provides an estimate of the surface spectral reflectance for each band as it would have been measured at ground level if there were no atmospheric scattering or absorption. Corrections are applied to compensate for the effects of the atmospheric gases and aerosols (Vermote, 2015a; Vermote, 2015b). In order to resolve surface reflectance with near-nadir illumination and nadir viewing angles, the MODIS MCD43C1 version 6 bidirectional reflectance distribution function (BRDF) and albedo parameters are

used. The BRDF describes how the reflectance depends on view and solar angles. Specification of the BRDF provides land surface reflectance explicitly in terms of its spectral, directional, spatial, and temporal characteristics (Lucht et al., 2000). The operational MODIS BRDF/albedo algorithm makes use of a kernel-driven, linear BRDF model, which relies on the weighted sum of an isotropic parameter and two functions (or kernels) of viewing and illumination geometry to determine reflectance as (Schaaf et al., 2002)

$R(\theta, \upsilon, \phi, \lambda) = f_{iso}(\lambda) + f_{vol}(\lambda) K_{vol}(\theta, \upsilon, \phi, \lambda) + f_{geo}(\lambda) K_{geo}(\theta, \upsilon, \phi, \lambda)$ ,                    (1)



where θ, υ, ϕ, λ are the solar zenith, view zenith, relative azimuth angles and wavelength, respectively. $K_k(\theta, \upsilon, \phi, \lambda)$ are the model kernels and $f_k(\lambda)$ are the spectrally dependent BRDF kernel parameters.

The MODIS surface reflectance used in Section 3 is calculated by Eq. (1) with $f_k(\lambda)$ from MODIS MCD43C1 provided BRDF/albedo parameters at band 4, and the model kernels $K_k(\theta, \upsilon, \phi, \lambda)$ by Eq. (38) and (39) of (Lucht et al., 2000) with 3°

solar zenith and view zenith angle and 0° relative azimuth angle. The MODIS MCD43C1 products are publicly available and can be downloaded from NASA's Land Processes Distributed Active Archive Center (LP DAAC).

**2.3 ICESat/GLAS surface elevation**

NASA Ice, Cloud and land Elevation Satellite (ICESat), which operated between 2003 and 2009, made the first satellite-based global nadir lidar measurements, highlighted by observations of Earth's ice sheet elevations, sea-ice thickness, and

vegetation canopy structure. The primary instrument on ICESat was the Geoscience Laser Altimetry System (GLAS), which utilized 1064 nm laser pulses for measuring the heights of the surface and samples the Earth's surface from an orbit of about 600 km with laser footprints of about 65 m in diameter spaced at 172 m along track. The primary objective of the GLAS laser altimeter was to measure elevation profiles of the Greenland and Antarctic ice sheet surfaces (Zwally et al., 2002). The GLAS/ICESat 500 m laser altimetry Digital Elevation Model (DEM) product of Antarctic (DiMarzio, 2007) is used in this

paper. This DEM is generated from the first seven operational periods (from February 2003 through June 2005) of the GLAS instrument. It is provided on polar stereographic grids at 500 m grid spacing. The grid covers all of Antarctica north of 86° S.

**3 CALIOP laser pulse bi-directional reflectance**

In this section, we introduce the surface laser pulse bi-directional reflectance retrieval method. First, we introduce the CALIOP receiver impulse response, which is critically important for analyzing the surface signals. Then we describe our

technique for recovering the saturated signal from snow and ice surfaces based on information gleaned from the subsurface "tail". The accuracy of the CALIOP surface laser pulse bi-directional reflectance results are evaluated using internal consistency checks and comparisons to MODIS.

**3.1 Method**

The detectors for the respective linearly orthogonal polarized channels of the CALIOP 532 nm receiver are photomultiplier

tubes (PMTs) that are coupled to third-order low-pass Bessel filters having a bandwidth ~2.44 MHz (Venkata and Reagan, 2016). The outputs of the low-pass filters are sampled by the 10 MHz analog-to-digital converters (ADC), which yield an equivalent 15 m vertical resolution. To conserve downlink bandwidth, the data acquired between 0.5 km below and 8.3 km above mean sea level are subsequently averaged on board the satellite to 30 m vertical resolution. A feature of the low-pass



Bessel filter is that for a short pulse return (as occurs from a hard-target reflection) the area under the pulse is preserved. Figure 1(a) illustrates a simulated example where the black curve is the hard surface return, the blue curve is the impulse response of the low-pass filter and the red curve is the convolution results between the black and blue curves. The surface elevation is assumed to be at zero meters.

It can be seen from Fig. 1(a) that if a narrow pulse (black) enters the low-pass filter, its peak amplitude is reduced but the pulse is stretched in time (or range r=ct/2, c the light speed, t travelling time) so as to preserve the area under the original narrow pulse. The shape of the post filter response is the CALIOP impulse response function, which is the convolution of the original narrow pulse and the impulse response of the low-pass filter. Because the width of the filter impulse response is much larger than the surface pulse, the shape of CALIOP impulse response function is dominated by the filter impulse

response and is relatively less sensitive to the pulse width of the transmitted laser pulse (20 ns width). Note that the red curve of Fig. 1(a) completely overlaps (and thus totally obscures) the blue curve.

Figure 1(b) shows the CALIOP impulse response model for the hard surface (blue curve, Hu et al., 2007) and simulated peak-normalized surface pulse (black). Since the original narrow surface pulse of Fig. 1(a) is stretched to the red line, the low pass filter can distribute a narrow surface return (black line) to a sequence of downlinked 30 m resolution samples (red

circles) over several adjacent range bins starting from the bin that contains the surface echo. Note that the downlinked samples in Fig. 1(b) have no time delay (Venkata and Reagan, 2016); that is, the samples digitized by the ADC are exactly coincident with the surface return pulse. However on board CALIOP, the start of the ADC sampling time varies, and thus no single sample is guaranteed to be exactly coincident with the Earth's surface, and the peak measured signal may be offset from the true surface altitude by several meters.

Figure 2 shows examples of the downlinked samples measured for different ADC start times. The surface altitude ($z_{surface}$) is assumed to be at zero meters (red dashed line). The black (Fig. 2a) and green (Fig. 2b) dashed lines show the elevations of the maximum value of the downlinked surface samples ($z_{peak}$). The elevation of downlinked surface peak of Fig. 2(a) is lower than the true surface elevation and simultaneously the magnitude of samples immediately before the peak is increased and the magnitude of samples immediately after the peak is reduced. When the elevation of downlinked surface peak is

higher than the surface elevation, the corresponding changes for the downlinked samples immediately before and after the peak are reversed (Fig. 2(b)). Figure 2 indicates that the shape of the downlinked samples' distribution is determined by the distance between actual surface elevation and the elevation of downlinked surface peak ($z_{surface}$-$z_{peak}$) (Lu et al., 2014).

The observations show that the CALIOP impulse response exhibits a non-ideal recovery of the lidar signal after a strong backscattering target. Examples of strong targets are dense liquid water clouds and surface returns (Hu et al., 2007; Lu et al.,

2014). As a result, the surface total integrated attenuated backscatter (unit: $sr^{-1}$) is calculated as integrating between 30 meters above and 300 meters below the surface as

$$\gamma'_{total} = \int_{surface\text{-}300m}^{surface+30m} \beta'(z)dz \,, \tag{2}$$



with the surface attenuated backscatter coefficients, $\beta'(z)$, obtained directly from CALIOP V4 L1 data product. The surface is determined to be the maximum attenuated backscatter located within ±150 meters of the Digital Elevation Model (DEM) surface altitude given by the 'Surface_Elevation' parameter recorded in the CALIOP L1 data.

Because of the non-ideal recovery of transient impulse response, the downlinked samples of Fig. 1(b) wrongly appear as if the laser pulse is penetrating the surface to a depth of several hundreds of meters. Comprehensive analyses by the CALIPSO team have determined that more than 90% of the surface return energy is contained in the first three downlinked samples, corresponding to the peak sample as well as one before and one after the peak sample as shown in Figs. 1(b) and 2. The remaining surface return energy is distributed into the signal "tail" that lies below the three peak samples (Hu et al. 2007). The tail signal is assumed to be extinguished at a distance of 300 m below the peak, so that the integrated attenuated backscatter (unit: sr$^{-1}$) of the surface tail is calculated as

$$\gamma'_{tail} = \int_{surface\text{-}300m}^{surface\text{-}60m} \beta'(z)dz \ , \tag{3}$$

The observed CALIOP surface total integrated attenuated backscatter is the product of the actual surface bidirectional reflectance and atmospheric effective transmission (Josset et al., 2010): $\gamma'_{total} = \rho T^2/\pi$. The land surface bi-directional reflectance ($\rho$) can be obtained directly from the surface total integrated attenuated backscatter as

$$\rho = \gamma'_{total}\pi/T^2 \ , \tag{4}$$

Here, T$^2$ is the two-way atmospheric transmittance, which can be estimated from the CALIOP data (Hu et al., 2008; Young and Vaughan, 2009). For example for the clear sky, the molecular Rayleigh two-way atmospheric transmittance can be estimated directly from Rayleigh extinction cross-section and molecular number density reported for each lidar profile from the ancillary meteorological data provided by the NASA Global Modeling and Assimilation Office (GMAO) (https://gmao.gsfc.nasa.gov/operations/). For cloudy (or aerosol) skies, the total column optical depth used to estimate the two-way atmospheric transmittance can be found from CALIOP L2 layer or profile products. The ocean surface reflectance from CALIOP lidar measurements can be found in detail from reference (Venkata and Reagan, 2016). However, the signals from snow and ice surfaces under clear skies are so strong that they usually saturate the digitizers. For the saturated signal, we should recover the saturated signal first before calculating the surface bi-directional reflectance from Eq. (4).

## 3.2 Recovery of CALIOP Saturated signal

Two 14-bit ADCs with different gains are used in each 532-nm channel to provide a 22-bit effective dynamic range (Hunt et al., 2009). On each channel, the high-gain ADC measures weak signals and the low-gain ADC acquires signals that saturate the high-gain digitizer. The profile samples are taken from the high-gain ADC if they are on-scale. If a sample is saturated on the high-gain ADC, the corresponding sample from the low-gain ADC, rescaled by the gain ratio between the high-gain channel and the low-gain channel (high-low gain ratio), is used. The output of each pair of digitizers is re-scaled and merged





into a single profile before being downlinked. The downlinked signal is saturated when the low-gain channel signal is saturated. The value that gets rescaled is the backscatter signal from the low-gain channel, which is the total signal minus the offset.

The maximum digitizer reading is 16383 (14 bits). Thus, the maximum backscatter value when the low-gain channel reaches

saturation is 16383-offset. The corresponding saturation value is the maximum low-gain channel backscatter (16383-offset) multiplied by the high-low gain ratio. Moreover, because multiple samples are averaged before downlinking, it is possible that a downlinked sample value might be less than the saturation value, and yet one of the samples that went into the average might be saturated. Note that the saturation threshold described here cannot be applied directly to the attenuated backscatter coefficients reported in the CALIOP L1 data product. To apply the saturation calculation to L1 data, some additional

conversions (e.g., range-scaled, energy normalized, calibration and amplifier gain normalized, etc.) are required.

For user convenience, a surface saturation flag used to indicate the likelihood that the surface backscatter signal is saturated has been added to the CALIOP V4.1 L1 data product. Surface saturation flag values of 0, 1 and 2 mean the surface backscatter signal is not saturated, possibly saturated and certainly saturated, respectively. In order to see the saturated and not saturated surface backscatter signal, Figs 3 and 4 show the relation between total and tail surface integrated attenuated

backscatter in the CALIOP 532 nm parallel and perpendicular channels, respectively. The colors represent the number of CALIOP lidar observations. Figure 3(a) shows all surface observations for the 532 nm parallel channel. Similarly, Fig. 3(b) shows only those surface observations that are not saturated, Fig. 3(c) shows the observations that are possibly saturated, and Fig. 3(d) shows only those parallel channel surface observations that are identified as 'certainly saturated'.

The green and black lines are the simulated total-to-tail signal ratios for not-saturated and saturated surface backscatter

signals from the CALIOP impulse response model shown in Fig. 1(b), where the saturation value is set to be 1.4 $km^{-1}sr^{-1}$ and the simulated downlinked samples have no time delay. The green and black lines overlap when the surface returns are small and not saturated. However, the black line departs significantly from the green line when the surface returns are saturated. Figure 4 is the same as Fig. 3 but for the CALIOP 532 nm perpendicular channel. Figures 3 and 4 show that there is a linear relation between total and tail surface-integrated attenuated backscatter (green line) for not-saturated surface backscatter

signals.

For the saturated signals, the total integrated attenuated backscatter ($\gamma'_{total}$) can be estimated from the surface tail $\gamma'_{tail}$ as

$$\gamma'_{total} = c\gamma'_{tail} , \tag{5}$$

The value of c is the total-to-tail signal ratio, which can be empirically estimated from fitting the total and tail signals of not-saturated CALIOP observations shown in Figs. 3(b) and 4(b), and/or theoretically obtained from the CALIOP impulse

response model shown in Fig. 1(b). The value of c is about 19.6±3.5 from fitting the not-saturated signals. Since the shape of the downlinked sample distribution is determined by the distance between actual surface elevation and the elevation of downlinked surface peak ($z_{surface}-z_{peak}$), the value of total-to-tail signal ratio is also a function of the distance $z_{surface}-z_{peak}$ as shown in Fig. 5, which is estimated from the impulse response model of Fig. 1(b) by setting different ADC sampling times





corresponding to the distance $z_{surface}$-$z_{peak}$ from about -15 m to 15 m. The advantage of using the surface tail to estimate the total integrated attenuated backscatter (Eq. 5) is that it will not be subject to saturation.

**3.3 Performance Assessment**

The surface return from snow and ice would saturate the detectors under clear sky or optically-thin cloud situations. Figure 6

shows the CALIOP data image of 532 nm total attenuated backscatter ($km^{-1}$ $sr^{-1}$) on 10 October 2009 over Antarctic snow surface. The color bar on the right indicates the magnitude of the total attenuated backscatter ($km^{-1}$ $sr^{-1}$) at 532 nm. The surface returns under clear sky within the red dashed lines are certainly saturated, while the surface return signals between the green dashed lines under transparent cloud are not. The surface signals lying underneath the transparent cloud layer between the rightmost red line and the leftmost green line are possibly saturated. The 532 nm non-ideal transient recovery is

seen in Fig. 6 as a gradual transition of colors from high attenuated backscatter values (white) to lower ones for the snow surface under clear sky (between the red dashed lines). The CALIOP ground-track orbit is shown as an intermittent black line in Fig. 7, with the interspersed green line segments corresponding to the not-saturated region in Fig. 6, and red line segments corresponding to the saturated region. The background color is the surface elevation (units of meters) from GLAS/ICESat 500 m laser altimetry Digital Elevation Model (DEM) product of Antarctica (DiMarzio, 2007).

For the saturated region of Fig. 6, the total integrated attenuated backscatter ($\gamma'_{total}$) of the snow surface is calculated from the surface tail by Eq. (5), with the total-to-tail signal ratio obtained from the relationship plotted in Fig. 5. The required elevation difference ($z_{surface}$-$z_{peak}$) is found between the GLAS/ICESat Antarctica DEM and CALIOP downlinked surface peak. Note that both the GLAS/ICESat Antarctica DEM and CALIOP elevation refer to the EGM96 Geoid datum. Here the GLAS/ICESat Antarctica DEM is assumed to be the true surface elevation. The surface bi-directional reflectance ($\rho$) is then

obtained from Eq. (4) with the clear sky two-way atmospheric transmittance derived from meteorological data reported in the CALIOP L1 data.

The surface bi-directional reflectance estimated from the surface tail is presented in red of Fig. 8(a). For comparison, the CALIOP observed surface bi-directional reflectance, with the total integrated attenuated backscatter ($\gamma'_{total}$) calculated by Eq. (2) directly from saturated surface return, is shown in blue. For the not-saturated region of Fig. 6, the cloud optical depth

($\tau_c$) and effective two-way transmittance ($T^2$) come from the corresponding CALIOP L2 cloud layer data products. However, when the cloud effective two-way transmittance is not available from L2 data, it is estimated as $\exp(-2\tau_c)(1+\tau_c/2)^2$ (Yang et al., 2013). The mean relative difference of effective two-way transmittances between first and second scattering orders is less than 10% when cloud optical depth is less than 1 (Yang et al., 2013).

The CALIOP observed surface reflectance before and after correcting for the cloud transmittance are shown in blue and red

in Fig. 8(b), respectively. The pink in Fig. 8 are the corresponding MODIS surface reflectances at 555 nm on 10 October 2009. There are no MODIS results over the left part of Fig. 8(a). Figure 8 indicates that the surface reflectances calculated





directly from the saturated signals (blue in Fig. 8(a)) will be, as expected, lower than the true values. Moreover, when clouds are present, the surface reflectances calculated before correcting for the cloud two-way transmittance (0.18±0.04) are much smaller than the reflectances calculated from the saturated signal under clear sky conditions (0.58±0.02). The mean surface reflectance estimated from the surface tail over the saturated region is about 0.86±0.06, while the mean surface reflectance

by correcting the cloud transmittance from not-saturated region is about 0.89±0.07. The MODIS surface reflectances are about 0.93±0.04 and 0.92±0.04 for the saturation and cloud regions, respectively. Note that almost all of the MODIS BRDF quality in Fig. 8 are large than 0, where 0 means the best quality of the MODIS data. The mean surface reflectance estimated from the surface tail is consistent with the reflectance after correcting the cloud transmittance and MODIS reflectance.

Figure 9 presents more CALIOP surface reflectance comparisons over Antarctica for clear sky (solid curves) and cloudy sky

(dashed curves) conditions in 2009. The solid black is the reflectance distribution estimated from the surface tail with the constant total-to-tail signal ratio of 19.6 used in Eq. (5), while the solid red is reflectance directly obtained from the saturated signals by Eqs. (2) and (4). The dashed red and black are surface reflectances before and after correcting for the cloud transmittance. The clouds are chosen with an optical depth of about 1 to make sure that the surface return under the cloud is not saturated and still exhibits a reasonably robust signal-to-noise ratio.

When clouds are present, the apparent mean surface reflectance before correcting the cloud transmittance (red dashed curve) is about 0.26±0.05. The apparent mean surface reflectance from saturated signal is about 0.58±0.05. The red dashed distribution is totally separated from the solid red distribution, which indicates that the apparent surface reflectance at 532 nm over polar snow/ice sheet regions could be used as cloud screening when cloud optical depth is greater than 1. The mean surface reflectance estimated from the surface tail (solid black) is about 0.90±0.10, while the mean surface reflectance

estimated by correcting for the cloud transmittance (dashed black) from the cloudy region is about 0.84±0.13. The surface reflectance under cloudy conditions will be more accurate if the cloud effective two-way transmittance can be obtained more accurately. The pink is MODIS reflectance distribution at 555 nm with the mean of about 0.96±0.04 over Antarctica in 2009. The surface bidirectional reflectance results shown in Fig. 9 are from one-year observations over Antarctica. Therefore, the surface reflectance might be variable due to snow aging, dust or soot content, melting and surface roughness etc. As a

consequence, the standard deviation of the surface reflectance reported here is larger than the simulated one of about 0.004 (Yang et al., 2013). For example, we assume here a 5% uncertainty in $\gamma'_{tail}$ and two-way atmospheric transmittance $T^2$, and 10% uncertainty in the total-to-tail signal ratio c. From Eqs. (5) and (4), the uncertainty of surface bi-directional reflectance can be estimated as $\frac{\Delta^2 \rho}{\rho^2} = \frac{\Delta^2 \gamma'_{tail}}{\gamma'^2_{tail}} + \frac{\Delta^2 c}{c^2} + \frac{4\Delta^2 T^2}{T^2}$. Summing up all the uncertainties yields the value of $\frac{\Delta^2 \rho}{\rho^2}$ as 0.02, resulting in an uncertainty in surface bi-directional reflectance ($\frac{\Delta \rho}{\rho}$) of about 14%. The standard deviation will be about 0.13 with the

mean value of surface bi-directional reflectance of about 0.9. The retrieved CALIOP surface bidirectional reflectance from



surface tail is consistent with reflectance for snow samples with melting and refreezing cycle (Wiscombe and Warren, 1980), and agrees with reflectance for snow samples in presence of volcanic sand and soot (Peltoniemi et al., 2015).

Figure 10 shows the monthly CALIOP surface reflectance at 532 nm for clear sky conditions in 2009, color-coded according to the retrieved surface reflectance. The surface reflectances for the saturated observations are obtained from the surface

tails, with a constant total-to-tail signal ratio of 19.6, while the surface reflectance for not-saturated observations are estimated directly from Eq. (2) and (4). The monthly surface reflectance shows that the reflectance values over permanent snow and ice surfaces (Greenland and Antarctica) are high during the whole year, and the reflectance results over snow covered land surface in winter time are higher than those in summer time when the snow are completely melted. There is significant seasonal reflectance transition from winter (December, January and February), to spring (March, April and May),

to summer (June, July and August) and then to fall (September, October and November). The pattern of CALIOP surface reflectance at 532 nm is similar to that of MODIS surface reflectance at 555 nm presented in Fig. 11, which is calculated from MODIS provided BRDF/albedo parameters with 3° off nadir illumination angles. We note that there are no MODIS results at high latitude during polar night. This new surface reflectance records from CALIOP, which makes reliable measurements both day and night and has an ability to characterize the surface even when obscured by transparent cloud/or

aerosol, complement existing MODIS passive remote sensing capabilities.

## 4 CALIOP Column bi-directional reflectance

In addition to the surface bidirectional reflectance work, a further effort is underway to improve the accuracy of the column top-of-atmosphere bi-directional reflectances currently calculated using the CALIOP lidar background data, according to the equation

$$\rho = \frac{\pi I}{\mu_0 S_0 D},\tag{6}$$

where $\mu_0$ is cosine of the solar zenith angle, and $I$ is the upwelling radiance derived from the lidar background signals. The upwelling radiance is proportional to the lidar background, which is proportional to the square of the root mean square (RMS) noise. $S_0$ is the extraterrestrial solar irradiance over the lidar bandpass. $D$ is an adjustment for the correct Earth-Sun distance (Spencer, 1971). The upwelling background radiance, $I$, can be derived from either the background monitor anode

current readings or the RMS baseline noise. The background monitor readings may be subject to uncertainty in offsets and also the possibility of saturation over bright targets. The square of the RMS baseline noise is highly correlated with the background monitor readings, but will not be subject to offset uncertainty. The RMS background noise is one of the downlinked status parameters and will not have to be computed from science data.

In terms of the RMS background noise, the parallel and perpendicular components of the upwelling background radiance can

be expressed as $I_{\parallel} = C_{\parallel} RMS_{\parallel}^2, I_{\perp} = C_{\perp} RMS_{\perp}^2$, where $RMS_{\parallel}$ and $RMS_{\perp}$ are the parallel and perpendicular components of the



baseline noise and $C_\parallel$ and $C_\perp$ are the parallel and perpendicular calibration coefficients that relate the RMS noise values to radiance units. In addition, the calibration coefficients are related through the lidar polarization gain ratio (PGR) as $C_\perp = PGR \times C_\parallel$. The parallel and perpendicular components of the column reflectance can be expressed as,

$$\rho_\parallel = \frac{C_\parallel RMS_\parallel^2}{\mu_0 S_0 D}, \tag{7}$$

$$\rho_\perp = \frac{PGR \times C_\parallel RMS_\perp^2}{\mu_0 S_0 D}, \tag{8}$$

The total column reflectance is then the sum of parallel and perpendicular components.

The CALIOP L1 data product provides the values of parallel and perpendicular column reflectance at single laser shot resolution. Figures 12, 13 and 14 show examples of column reflectance from L1 data on 4 September 2006. The blue line of Fig. 12 is the CALIOP ground orbit track where the red one is the region for the results shown in Figs. 13 and 14. CALIOP

532 nm total attenuated backscatter (km$^{-1}$ sr$^{-1}$) is given in Fig. 13, which shows the cloud and ocean surfaces. In Fig. 14 we show the column reflectance values from both the CALIOP lidar (red) and the WFC (green) for the orbit shown in red line of Fig. 12. The column reflectance values computed from the lidar data are consistent with the values from the camera. The relative differences are within 20%. The reason for the discrepancies could come from calibration biases, surface FOV differences and detector spectral response function differences. A global map of CALIOP column reflectance under clear sky

condition for each month in 2009 is shown in Fig. 15. The column reflectance is directly from CALIOP level 1 data and averaged over 1° by 2° latitude and longitude grid boxes. The column reflectances for Greenland and Antarctica are seen to be fairly consistent values in the range from 0.8 to about 1.1 with a mean value of 0.94±0.10. The column reflectance over ocean surface varies between near zero to about 0.2. For the land surfaces, the column reflectance can be from 0.05 to about 0.4, with somewhat higher values in desert regions. Note that the column reflectance is available only on CALIOP daytime

measurements since it is retrieved from the solar background signals. The CALIOP column bi-directional reflectance records can be used to model the solar background radiance in the lidar simulator tool (Powell et al., 2017).

**5 Conclusion and Summary**

In this paper, we developed an innovative retrieval method that translates the Cloud-Aerosol Lidar with Orthogonal Polarization (CALIOP) land surface laser pulse returns into the surface bidirectional reflectance. The method takes

advantage of the CALIOP impulse response, which distributes a narrow surface return to a sequence of downlinked 30 m resolution samples over several adjacent range bins. The CALIPSO team have determined that more than 90% of the surface return energy is contained in the first three 30-meter vertical range bins starting from the bin that contains the surface echo, and the remaining surface return energy is distributed into the surface "tail" that lies below the first three range bins. This new method takes advantage of surface tail to recover the saturated pulse return from snow and ice surfaces. In CALIOP



V4.1 L1 data product, surface saturation flag values of 0, 1 and 2 mean the surface backscatter signal is not-saturated, possibly saturated and certainly saturated, respectively. For not-saturated surface backscatter signals, the total attenuated backscatter ($\gamma'_{total}$) was calculated as integrating of the surface attenuated backscatter coefficients between 30 meters above and 300 meters below the surface. For the possibly and certainly saturated signals, the total attenuated backscatter ($\gamma'_{total}$)

can be estimated from the surface tail with an empirically estimated total-to-tail signal ratio. The land surface bi-directional reflectance ($\rho$) can be finally obtained from the total attenuated backscatter ($\gamma'_{total}$) with the effective two-way atmospheric transmittance estimated from CALIOP data.

To validate the new method, the snow surface bidirectional reflectances obtained from the tail of the surface signals were compared with Moderate-resolution Imaging Spectroradiometer (MODIS) surface reflectances and surface reflectances

calculated from CALIOP not-saturated surface returns when cloud is present. The comparisons show that the snow surface bidirectional reflectances over Antarctica for saturation region are generally reliable, with a mean value of 0.90±0.10, while the mean surface reflectance from cloud cover region is about 0.84±0.13 and the calculated MODIS reflectance at 555 nm from BRDF/albedo model with near nadir illumination and viewing angles is about 0.96±0.04. The comparisons here demonstrate that the apparent reflectance of snow surfaces beneath clouds with cloud optical depths of about 1 is

significantly lower than that for clear sky conditions, thus supporting the use of apparent surface reflectance in cloud detection/screening in satellite data analysis.

The column top-of-atmosphere bidirectional reflectance from CALIOP version 4 data products is calculated from the square of the RMS baseline noise. The RMS baseline noise is highly correlated with the CALIOP background monitor readings and is not subject to offset uncertainty or saturation. The comparison analysis shows that the column reflectance values computed

from the lidar data are consistent with the values from the WFC camera. For clear skies, the column bidirectional reflectance over Greenland and Antarctica is in the range from 0.8 to about 1.1 with mean value of 0.94±0.10, the column reflectance over ocean surface varies between near zero to about 0.2 and the column reflectance over land can be from 0.05 to about 0.4 with somewhat higher values in desert regions. These CALIOP column bi-directional reflectance records can be used to suggest the values of the solar background radiance required in the lidar simulator tools.

The CALIOP makes reliable and independent measurements both day and night, and at low solar angles through considerable aerosol loads and transparent clouds. The surface bidirectional reflectances retrieved from CALIOP measurements contribute complementary data for existing Moderate-resolution Imaging Spectroradiometer (MODIS) standard data products and could be used to detect and monitor seasonal surface reflectance changes in high latitude regions where passive MODIS measurements are limited. The more than 10 years of CALIOP continuous observations of snow

surface bidirectional reflectance and column top-of-atmosphere bidirectional reflectance in polar region will benefit the communities modelling snow melting and climate change. There is an ongoing study which uses the snow surface




bidirectional reflectances at 532 nm and 1064 nm and depolarization ratio at 532 nm for snow grain size and snow depth studies.

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



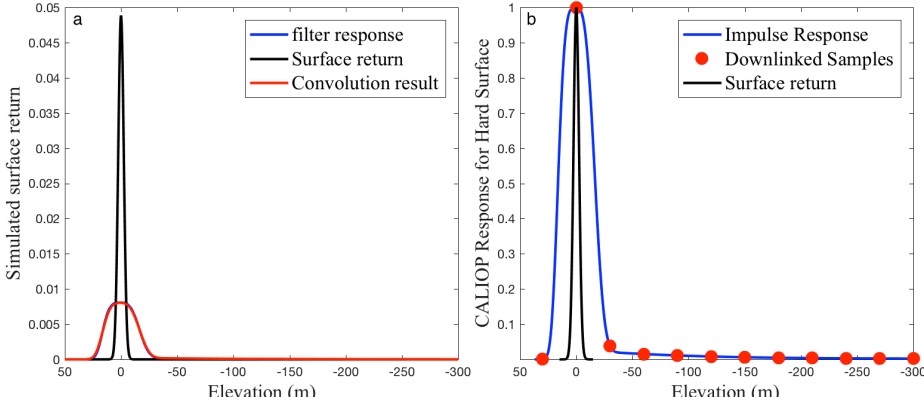

**Figure 1 (a) simulated low-pass filter response (blue), surface return (black) and the convolution result of the surface return and low-pass filter (red); note that the blue curve is completely overwritten by the red curve; (b) CALIOP impulse response function for hard surface (blue) [32], the downlinked 30 m vertical resolution samples (red circles) and the simulated peak-normalized surface return (black). Note that the downlinked samples in this example have no time delay [17], and the surface elevation is assumed to be at 0 meters. The elevation below the surface is set to a negative value.**

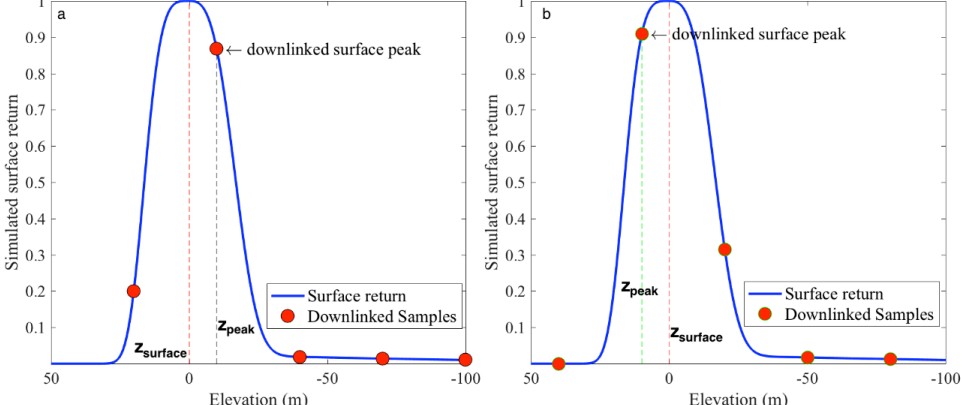

**Figure 2 The simulated surface return pulse after low-pass filter (blue curve) and CALIOP downlinked 30 m resolution samples (red circles). The surface elevation ($z_{surface}$) is assumed to be 0 meter (red dash line). The elevation below the surface is set to negative value. The black and green dash lines are elevations where the downlinked surface sample is peak ($z_{peak}$).**





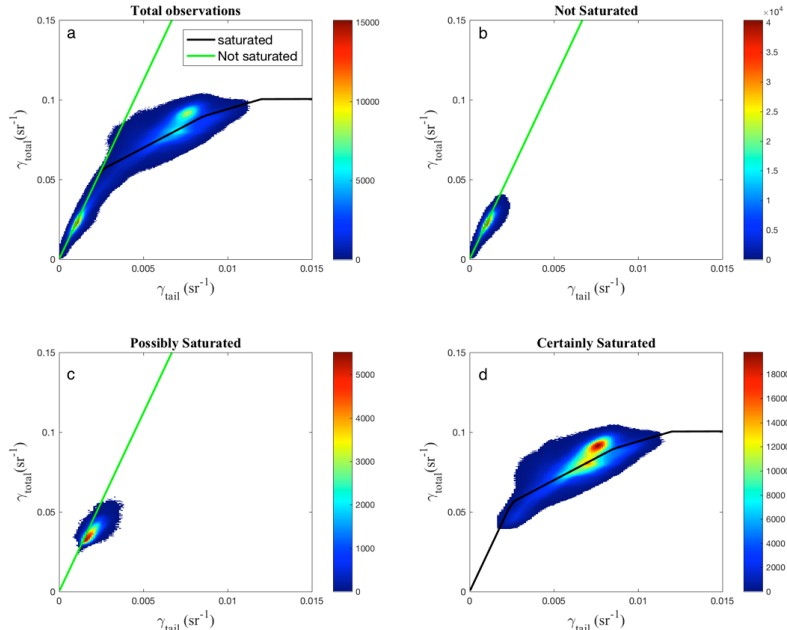

**Figure 3. Relation between total $\gamma'_{total}$ and tail $\gamma'_{tail}$ integrated attenuated backscatter at CALIOP 532 nm parallel channel. (a) total observations; (b) not saturated observations; (c) possibly saturated observations; (d) certainly saturated observations. The color bar is the number of observations. The green and black lines are the simulated total-to-tail signal ratios for not saturated and**
5 **saturated surface returns from the CALIOP impulse response function shown in Figure 1(b).**





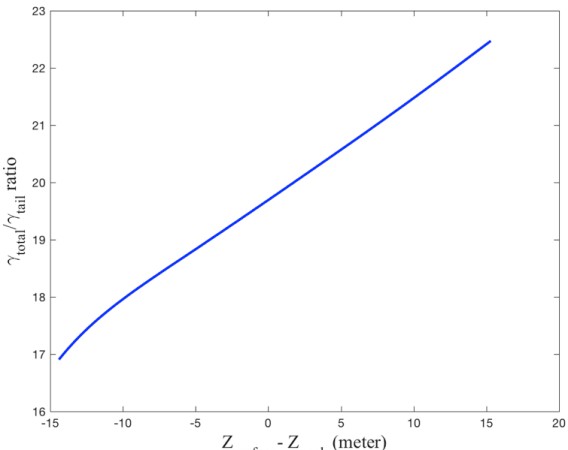

**Figure 4. The same as Figure 3 but for the CALIOP 532 nm perpendicular channel.**

**Figure 5** The total-to-tail signal ratio as a function of the elevation difference between actual surface and downlinked surface
5    sample peak $z_{surface}$-$z_{peak}$.



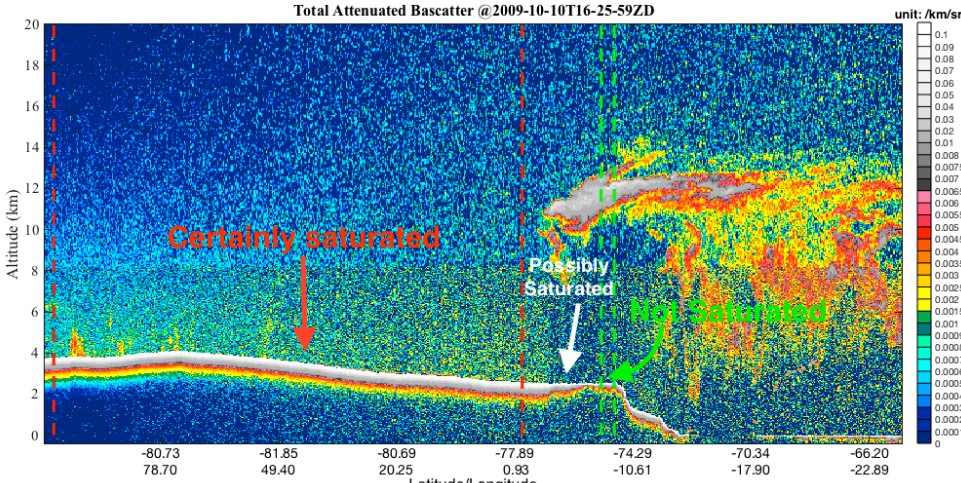

**Figure 6. Total Attenuated backscatter at 532 nm (km⁻¹ sr⁻¹) on 10 October 2009; The surface return signals within the red dash lines under clear sky are certainly saturated, while the signals between the green dash lines under transparent cloud are not saturated. The surface signals between red and green dash lines under transparent cloud are possibly saturated. The color bar on the right indicates the value of total attenuated backscatter.**

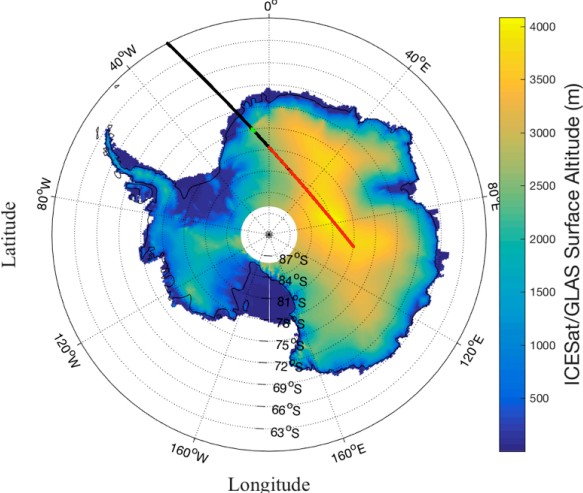

**Figure 7 CALIOP ground-track over Antarctic (black line). Green orbits are corresponding to the not saturated region and red orbits are corresponding to the saturated region of Figure 6; The background color is the snow surface elevation (unit meter) from GLAS/ICESat Antarctic 500 m DEM product.**





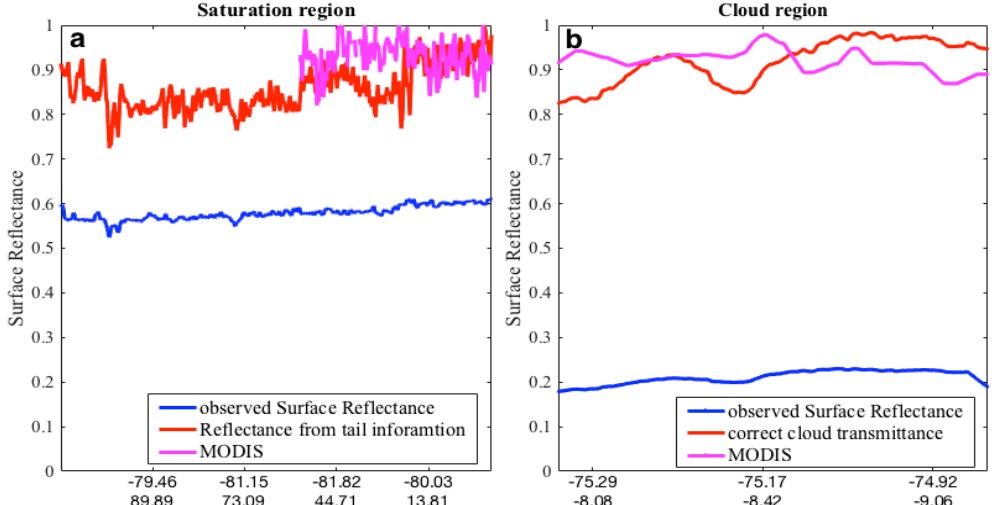

**Figure 8.** Left panel (a): CALIOP observed surface bidirectional reflectance (blue) and corrected surface bidirectional reflectance (red) corresponding to the saturated region of Fig. 6; Right panel (b): CALIOP observed surface bidirectional reflectance under transparent cloud before (blue) and after (red) correcting the cloud transmittance. It is corresponding to the not-saturated region of Fig. 6. The pink is MODIS surface reflectance at 555 nm.

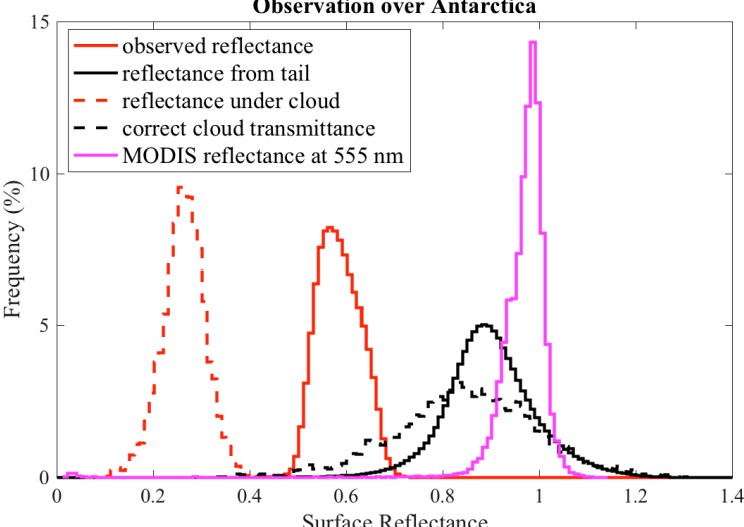

**Figure 9** The CALIOP surface bidirectional reflectance distribution at 532 nm over Antarctica for clear sky (solid curves) and cloud sky conditions (dashed curves). The solid red is reflectance directly from the saturated signals. The solid black is reflectance estimated from surface tail with the constant total-to-tail signal ratio of 19.6. The dash red and black are surface reflectance before and after correcting the cloud transmittance. The pink is MODIS reflectance distribution at 555 nm.





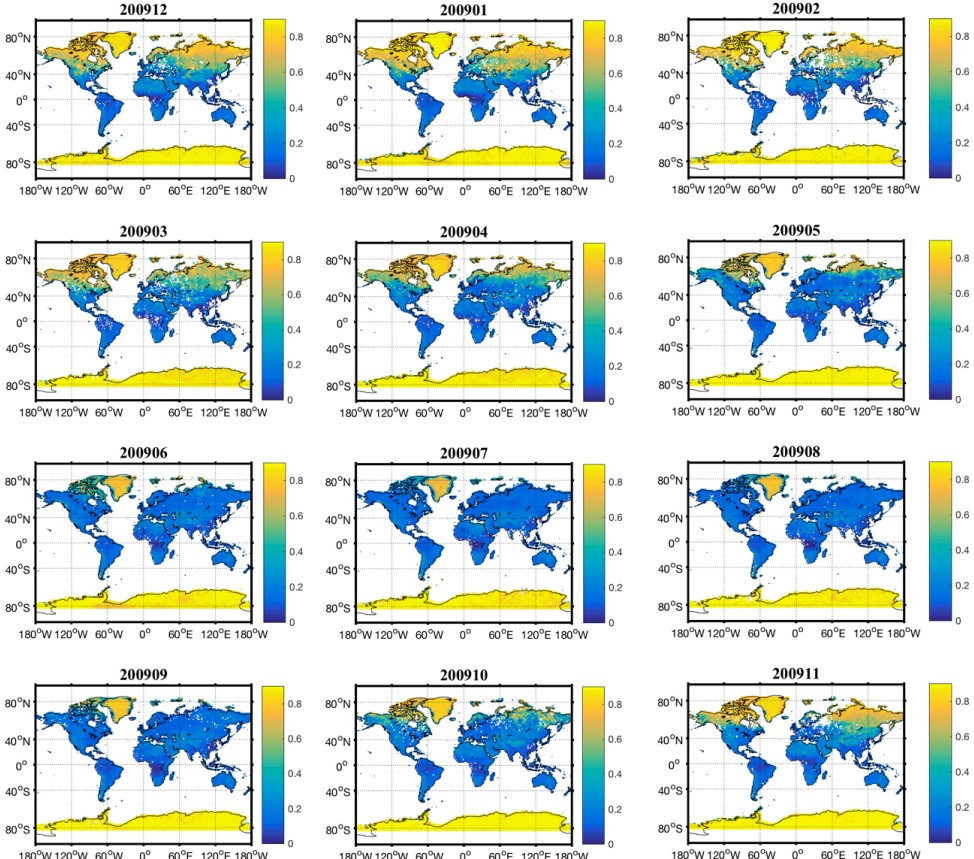

**Figure 10 The monthly CALIOP surface reflectance at 532 nm for clear sky condition in 2009. The colors represent the value of surface reflectance.**



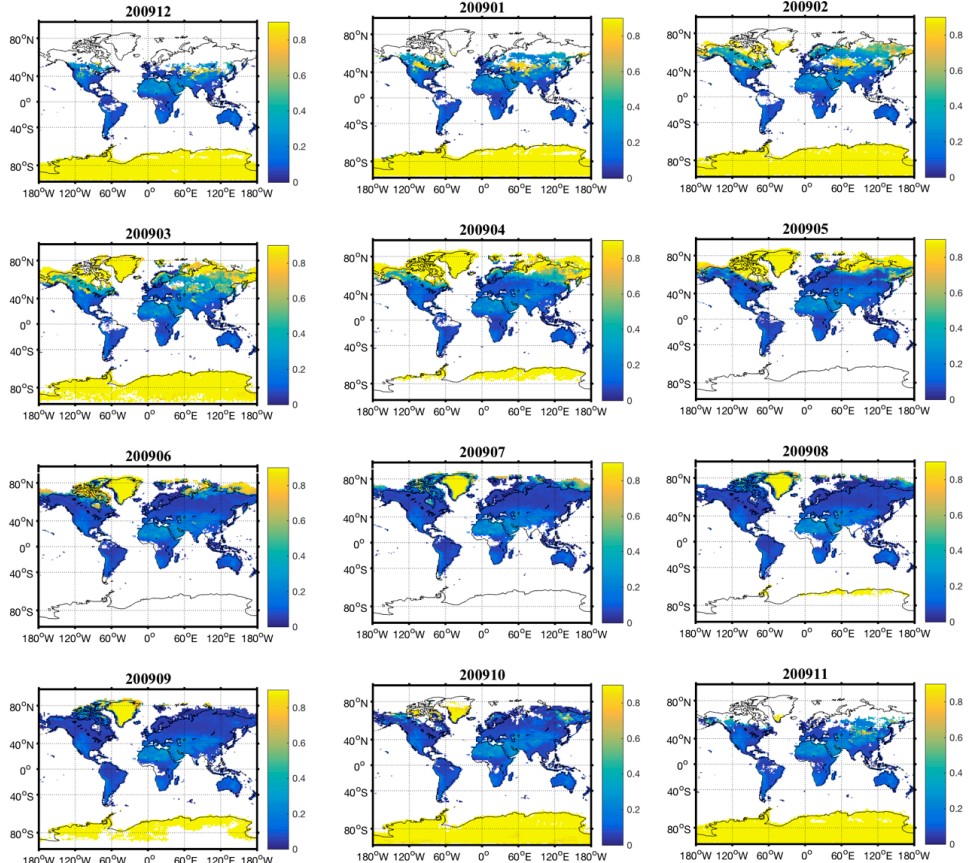

**Figure 11 The monthly MODIS surface reflectance at 555 nm in 2009. The colors represent the value of surface reflectance. There are no MODIS observations during polar night at high latitudes**





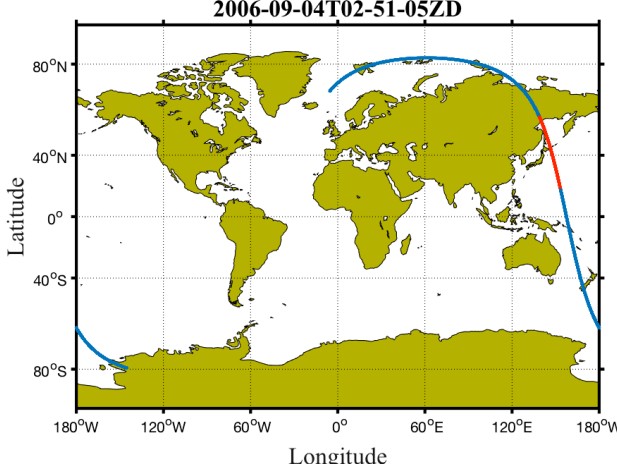

**Figure 12 CALIOP ground orbit track. The blue line is the orbit track of one granule around 2:51am on September 4th 2006. The red line is the study region shown in Figure 12 and 13.**

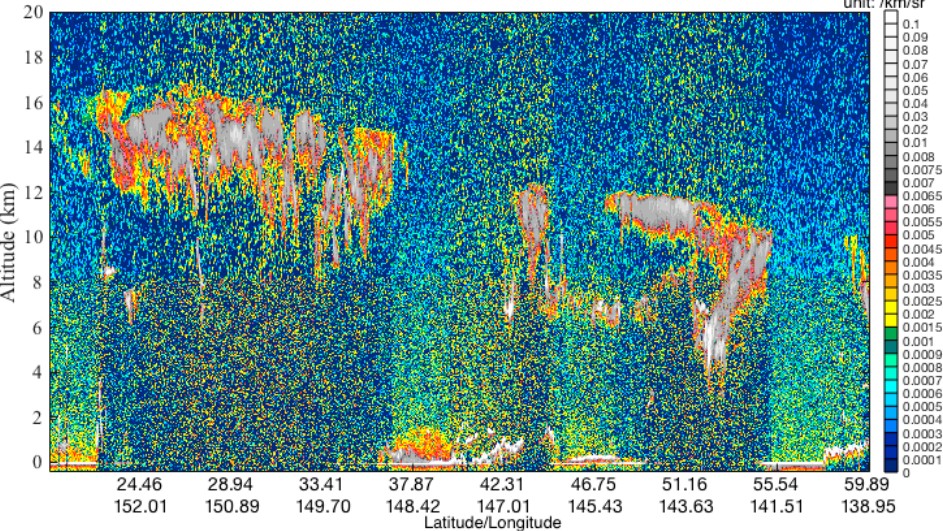

5   **Figure 13 Total attenuated backscatter (km⁻¹ sr⁻¹) along the red line orbit shown in Figure 11. The color bar on the right indicates the value of total attenuated backscatter.**



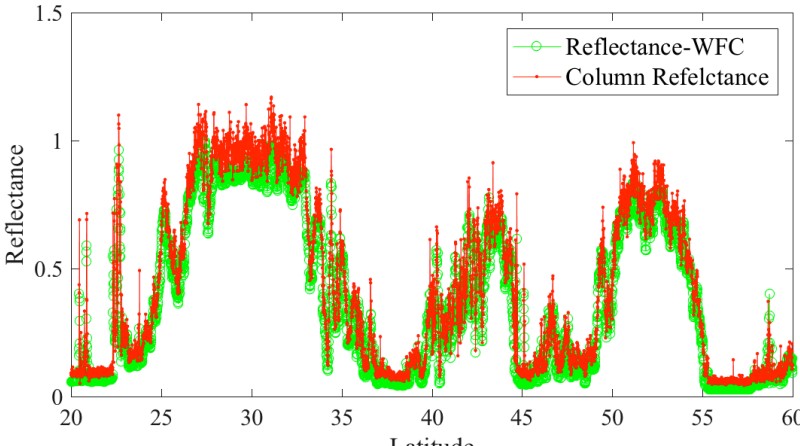

**Figure 14. CALIOP Column reflectance (red) compared with the WFC bi-directional reflectance (green).**





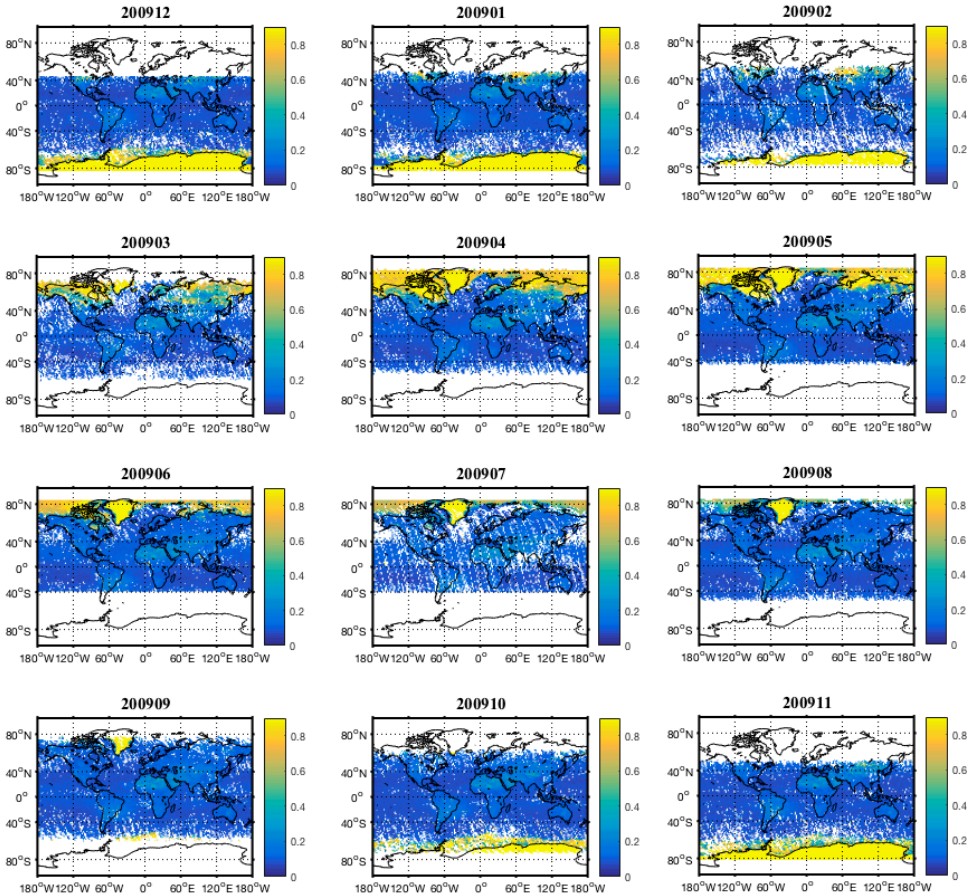

**Figure 15. The monthly CALIOP Column reflectance for clear sky condition in 2009. The areas of white are where there are insufficient observations to derive a meaningful value.**

