# Peer review of "Laser pulse bidirectional reflectance from CALIPSO mission"

_Atmospheric Measurement Techniques, 2017_

## Referee Comment (RC1) · Anonymous Referee #1 · 21 Feb 2018

This is a reasonably well written paper presenting significant results employing the CALIPSO lidar response from a hard surface. While others (e.g., Josset, et al., GRL, 2008, Venkata and Reagan, Remote Sens, 2016) have reported on retrievals of aerosol optical depth (AOD) from CALIPSO lidar ocean surface returns, this paper addresses recovering the surface retro reflectance (expressed in terms of the surface bi-directional reflectance, BRDF, which can be readily compared to MODIS retrieved BRDF'S) for snow and ice surfaces which present challenges in dealing with saturated signals for such bright targets. The authors present an innovative approach for identifying and recovering the saturated signals through use of both the parallel and perpendicular (depolarized) CALIPSO 532 nm lidar channels.

As the lidar surface response is proportional to the product of round-trip transmittance

to the surface times the surface reflectance, the reflectance can be recovered if the transmittance is known. The authors use the CALIPSO data product estimates of transmittance to recover the surface reflectance for quite clear (nearly Rayleigh) and thin cloud situations. Their retrieved reflectances (BRDF's for lidar backscatter), corrected for saturation or thin clouds, yield values in reasonable agreement with MODIS derived backscatter BRDF's, probably as good an agreement as possible given the likely uncertainty in the MODIS BRDF modeling for backscattering.

The authors define their recovered reflectance (eq. 4) in terms of the total observed surface backscattering signal which includes the long 'noise tail' observed in the lidar surface return (e.g., Hunt et al., JAOT, 2009). They could have alternately defined the reflectance in terms of the total minus tail (eq. 2 minus eq. 3) signal; i.e., main pulse signal. Which is a better/more correct is a matter of conjecture concerning whether the tail is true signal versus after-pulsing noise. The authors do note that over 90% of the surface return signal is contained in the main pulse portion, so either definition for the reflectance would yield about the same result.

In conclusion, this paper presents an innovative approach for recovering the surface BRDF (at backscatter) from CALIPSO surface return signals from snow and ice surfaces, even for saturated signal levels by using the parallel and perpendicular 532 nm lidar channels. The results are new and significant. The paper definitely merits publication.

---

## Referee Comment (RC2) · Anonymous Referee #2 · 9 Mar 2018

The paper presents in a clear and accurate way the application of a new methodology for the determination of bidirectional reflectivity of the solid earth surface, in particular from ice and snow.The proposed method uses the pulse produced by CALIOP on the surface to determine the bidirectional reflectance. The authors follow two approaches for the determination of reflectance, based respectively on the analysis of the entire response pulse or on the tail. The linearity of the reflectance dependence calculated with the first method with respect to that calculated with the second method justifies the use of the calculation made with the single tail, which is particularly relevant as it extends the application to the saturated signal cases. To validate the new method, the bidirectional surface reflectance obtained from the surface signal tail was success-fully compared with the data produced by the Moderate-resolution Imaging Spectro-

radiometer (MODIS) tool. The proposed method significantly integrates the current passive detection capabilities. In fact, MODIS does not provide data in high latitude areas during the polar night, while the proposed method provides reflectance data both day and night and also when the surface is obscured by transparent clouds or aerosol layers. The authors report the results of the determination of the reflectance of snowy surfaces under clouds with an optical depth of the cloud of about 1, in these cases the reflectance of the surface is systematically lower than that of the conditions of clear sky. This observation opens the field to the possibility of cloud screening starting from the measurement of surface reflectance. The article also describes the work aimed at improving the accuracy of the bidirectional high-atmosphere reflections using the background data of CALIOP. The data obtained in this application are compared with those of the reflectance measured with the Wide Field Camera (WFC) of CALIPSO. The article analyzes in detail the possible causes of disagreement, but overall the direct comparison is convincing. This article fully deserves to be published.

---

## Author Comment (AC1) · 19 Apr 2018

We would like to express sincere gratitude to the Reviewers for their careful reading and thorough comments. The step-by-step answers to the Reviewers' comments are listed below.

Referee #1

comments from Referees: This is a reasonably well written paper presenting significant results employing the CALIPSO lidar response from a hard surface. While others (e.g., Josset, et al., GRL, 2008, Venkata and Reagan, Remote Sens, 2016) have reported on retrievals of aerosol optical depth (AOD) from CALIPSO lidar ocean surface returns, this paper addresses recovering the surface retro reflectance (expressed in terms of

the surface bi-directional reflectance, BRDF, which can be readily compared to MODIS retrieved BRDF'S) for snow and ice surfaces which present challenges in dealing with saturated signals for such bright targets. The authors present an innovative approach for identifying and recovering the saturated signals through use of both the parallel and perpendicular (depolarized) CALIPSO 532 nm lidar channels.

Author's response: Yes, the paper presents an innovative approach to retrieve surface bidirectional reflectance from CALIOP surface laser pulse returns. The saturated signal from snow and ice surfaces are recovered based on surface tail for both CALIOP 532 nm parallel and perpendicular channels.

comments from Referees: As the lidar surface response is proportional to the product of round-trip transmittance to the surface times the surface reflectance, the reflectance can be recovered if the transmittance is known. The authors use the CALIPSO data product estimates of transmittance to recover the surface reflectance for quite clear (nearly Rayleigh) and thin cloud situations. Their retrieved reflectances (BRDF's for lidar backscatter), corrected for saturation or thin clouds, yield values in reasonable agreement with MODIS derived backscatter BRDF's, probably as good an agreement as possible given the likely uncertainty in the MODIS BRDF modeling for backscattering.

Author's response: Right, CALIOP level 2 data products provide cloud optical properties, such as cloud optical depth and two-way transmittance, and CALIOP level 1 data products provide Rayleigh extinction cross-section and molecular number density. As a result, the two-way atmospheric transmittance can be estimated from CALIOP level 1 and level 2 data products. Finally, the surface bi-directional reflectance is obtained from the surface total integrated attenuated backscatter with the known transmittance.

comments from Referees: The authors define their recovered reflectance (eq. 4) in terms of the total observed surface backscattering signal which includes the long 'noise tail' observed in the lidar surface return (e.g., Hunt et al., JAOT, 2009). They could have

alternately defined the reflectance in terms of the total minus tail (eq. 2 minus eq. 3) signal; i.e., main pulse signal. Which is a better/more correct is a matter of conjecture concerning whether the tail is true signal versus after-pulsing noise. The authors do note that over 90% of the surface return signal is contained in the main pulse portion, so either definition for the reflectance would yield about the same result.

Author's response: For an ideal detector, the surface signal will return immediately to its baseline state. However, because of the CALIOP non-ideal recovery of transient impulse response, the surface signal is stretched to a sequence of 30 m resolution bins starting from the bin that contains the surface echo. Comprehensive analyses by the CALIPSO team have determined that more than 90% of the surface return energy is contained in the first three bins, corresponding to the peak signal as well as one before and one after the peak signal. The remaining surface return energy is contained in the tail bins that lies below the three peak bins. As a result, the surface total integrated attenuated backscatter (Eq. 4) is used to obtain the surface bi-directional reflectance. The lidar signal from snow/ice surface is so strong that the signals at three peak bins are often saturated under conditions of clear sky. For the saturated lidar return, the surface total integrated attenuated backscatter is estimated from its surface tail (Eq. 5) which is not saturated.

comments from Referees: In conclusion, this paper presents an innovative approach for recovering the surface BRDF (at backscatter) from CALIPSO surface return signals from snow and ice surfaces, even for saturated signal levels by using the parallel and perpendicular 532 nm lidar channels. The results are new and significant. The paper definitely merits publication.

Author's response: Thanks;

---

## Author Comment (AC2) · 19 Apr 2018

We would like to express sincere gratitude to the Reviewers for their careful reading and thorough comments. The author's response to the Reviewers' comments are listed below.

Referee #2

comments from Referees: The paper presents in a clear and accurate way the application of a new methodology for the determination of bidirectional reflectivity of the solid earth surface, in particular from ice and snow. The proposed method uses the pulse produced by CALIOP on the surface to determine the bidirectional reflectance. The authors follow two approaches for the determination of reflectance, based respectively on the analysis of the entire response pulse or on the tail. The linearity of the reflectance dependence calculated with the first method with respect to that calculated with the second method justifies the use of the calculation made with the single tail, which is particularly relevant as it extends the application to the saturated signal cases. To validate the new method, the bidirectional surface reflectance obtained from the surface signal tail was successfully compared with the data produced by the Moderate-resolution Imaging Spectro-radiometer (MODIS) tool. The proposed method significantly integrates the current passive detection capabilities. In fact, MODIS does not provide data in high latitude areas during the polar night, while the proposed method provides reflectance data both day and night and also when the surface is obscured by transparent clouds or aerosol layers. The authors report the results of the determination of the reflectance of snowy surfaces under clouds with an optical depth of the cloud of about 1, in these cases the reflectance of the surface is systematically lower than that of the conditions of clear sky. This observation opens the field to the possibility of cloud screening starting from the measurement of surface reflectance. The article also describes the work aimed at improving the accuracy of the bidirectional high-atmosphere reflections using the background data of CALIOP. The data obtained in this application are compared with those of the reflectance measured with the Wide Field Camera (WFC) of CALIPSO. The article analyzes in detail the possible causes of disagreement, but overall the direct comparison is convincing. This article fully deserves to be published.

Author's response: The comment of this paper is right and correct. Thanks;

---

## Author Comment (AC3) · 19 Apr 2018

Author's response: We thank the reviewer for their very positive comments. We are especially gratified by the reviewer's recognition that, by using our new technique, CALIPSO is able to extend the measurement record to regions and times where MODIS retrievals are currently unavailable.
* * *

---

## Author Comment (AC4) · 19 Apr 2018

Author's response: Thank you. We very much appreciate the reviewer's endorsement.